# Enhanced Dielectric and Hydrophobic Properties of Poly(vinylidene fluoride-trifluoroethylene)/TiO_2_ Nanowire Arrays Composite Film Surface Modified by Electrospinning

**DOI:** 10.3390/polym13010105

**Published:** 2020-12-29

**Authors:** Da Shen, Qilong Zhang, Zhao Zhang, Hui Yang, Jiansong Sheng

**Affiliations:** 1School of Materials Science and Engineering, State Key Lab Silicon Mat, Zhejiang University, Hangzhou 310027, China; 21826004@zju.edu.cn (D.S.); 11926026@zju.edu.cn (Z.Z.); yanghui@zju.edu.cn (H.Y.); 2Research Institute of Zhejiang University-Taizhou, Taizhou 318000, China; shengjiansong@126.com

**Keywords:** TiO_2_ nanowire arrays, poly(vinylidene fluoride-trifluoroethylene), electrospinning, high permittivity, high contact angle

## Abstract

In this research, we designed a feasible method to prepare composite films with high permittivity and significantly enhanced hydrophobic performance, which showed huge potential in the electrowetting field. TiO_2_ nanowire arrays were prepared by a one-step hydrothermal process, and poly(vinylidene fluoride-trifluoroethylene) (P(VDF-TrFE)) was spin-coated on the nanowire arrays to form composite, the surface of which was modified by electrospinning. Due to the great orientation of TiO_2_ nanowires, dipoles and space charges are in ordered arrangement along the electric field, and this strongly reinforced the Maxwell–Wagner–Sillars (MWS) polarization, thus the permittivity of the composite (TiO_2_ nanowire length/film thickness is 0.769) reaches 53 at 1 kHz, which is nearly 3 times higher than pure P(VDF-TrFE). Meanwhile the composite film possesses low dielectric loss (0.07) and low conductivity (2.69 × 10^−9^ S/cm), showing good insulation. The contact angle of the composite after electrospinning (about 137°) was greatly enhanced from pure P(VDF-TrFE) spin-coated film (about 89°), which can be attributed to the microrough structure built by P(VDF-TrFE) nanofibers.

## 1. Introduction

The electrowetting phenomenon refers to the change in hydrophobicity when voltage is applied between the conductive droplet and surface. In order to avoid the electrolysis problem when the electric circuit is established through the droplet, a dielectric layer is added between the droplet and bottom electrode to build a plane-parallel capacitor, and the technique using dielectric layer in electrowetting is called electrowetting on dielectric (EWOD) [1,2,3,4]. As one of the most efficient methods to control the behavior of micro drops, EWOD has shown great potential in reflective display [5,6,7], lab on a chip [8,9], liquid lenses [10], etc.

Water droplets tend to form hemispheric or spherical shape on a hydrophobic surface because of the surface tension. The relationship between the contact angle of droplet and the surface tension of solid–liquid, solid–gas and gas–liquid interface can be described as the Young equation:
(1)$$cos\theta =\left(\gamma_{sg}-\gamma_{sl}\right)/\gamma_{lg}$$

Furthermore, according to the works reported by Lippmann on the electrocapillary phenomenon, when voltage is applied and charges appear on the solid surface, a coulombian force is established among charges both on the surface and inside the conductive droplets. The coulombian force partially counteracts the surface tension, and thus causes the decrease of the contact angle, as is shown in Figure 1. This becomes the mechanism of electrowetting, and is described as the Young–Lippmann equation [11,12]:(2)$$\cos\theta ={\cos\theta }_{0}+\;\frac{\varepsilon_{0}\varepsilon }{2{d\gamma }_{\mathrm{lg}}}V^{2}$$

It can be easily observed that the contact angle decreases when the voltage rises, so to achieve a wide variation range for contact angle, a higher initial contact angle (θ_0_, the contact angle without external voltage) is required. In addition, a lower voltage is needed for a same change in contact angle of the droplet when the dielectric constant ε is higher or the thickness d of the dielectric layer is lower, which means easier and more energy-saving electrowetting technology. As a result, the key research direction for EWOD is to acquire an insulting layer with both high dielectric constant and an initial contact angle.

To obtain such insulating films, various organic/inorganic materials have been used in electrowetting. Polymer materials, such as polymethyl methacrylate (PMMA) [13,14], polystyrene (PS) [15], polydimethylsiloxane (PDMS) [16] and Teflon AF [17,18] usually possesses relatively high initial contact angle, good flexibility but low dielectric. In some research, combining two or more types of polymers to build a pure organic system can be an effective way to promote the dielectric properties. For instance, Zhu et al. constructed a PDMS/P(VDF-TrFE) blend film with PS nanospheres as fillers. The pure inorganic composite showed excellent flexibility and markedly improved dielectric constant, about 21 with 23 wt % PS. Meanwhile the initial contact angle was increased from 110 to 125° because of the adding of PS nanospheres, realizing good dielectric properties and hydrophobicity within monolayer films [19].

Meanwhile, inorganic materials like ZnO [20,21], TiO_2_ [22,23], MgO [24] and BaTiO_3_ [25] show just the reverse characteristics. As a result, composing highly hydrophobic polymer matrix and inorganic fillers with good dielectric properties to acquire the advantages of both sides is another mainstream method [26]. Moreover, preparing inorganic fillers with specific microstructure to enhance the dielectric properties and build a rough microstructure on the surface to increase the initial contact angle are common ideas. For example, in the work of R. Magisetty et al., they fabricated ABS/NiFe_2_O_4_ composite films using Low Volume High-Pressure Air Atomization (LVHPAA) techniques. With 40 wt % NiFe_2_O_4_ added, the permittivity of the composite reached 10.5 at 10 Hz, extremely increased from pure ABS (2.4). In contrast to the films prepared by solution casting, the LVHPAA films showed higher permittivity, attributed to the greater dispersion of NiFe_2_O_4_. In addition, the composite obtained by LVHPAA showed higher surface roughness than solution casted films, thus possessed higher contact angle (123.3° with 10 wt % NiFe_2_O_4_ added, while the solution casted film was 101°) [27].

TiO_2_ has been widely used in EWOD composites because of its good dielectric properties, but the ratio of TiO_2_ fillers, such as nanoparticles or nanowires in the composites is limited owing to the agglomeration, thus limits the further increase of dielectric constant. Moreover, the orientation of TiO_2_ fillers can be improved to enhance the dielectric properties. As a result, in this research, we present a new material design for EWOD with selected growth of TiO_2_ nanowires. TiO_2_ nanowire arrays with excellent orientation along the vertical direction, prepared by a simple one-step hydrothermal reaction were composited as fillers with the ferroelectric β-phase P(VDF-TrFE) matrix. The surface of the composite film was treated by an electrospinning process to fabricate P(VDF-TrFE) nanofibers on it. The dielectric constant is enormously enhanced by the highly aligned TiO_2_ nanowire, achieving 53 at 1 kHz when the ratio of the TiO_2_ nanowire length and film thickness is 0.769. The dielectric loss is only about 0.07, and the conductivity is 2.69 × 10^−9^ S/cm, revealing good insulation. Meanwhile, the initial contact angle is tremendously increased from 89° (flat surface of P(VDF-TrFE)) to 137°, owing to the significant improvement of surface roughness caused by the P(VDF-TrFE) nanofibers.

## 2. Materials and Methods

### 2.1. Materials

Tetrabutyl titanate (≥99.0%) and *N, N-dimethylformamide* (99.5%) were purchased from Aladdin Industrial Corporation, Shanghai, China. HCl (37.0%) and acetone (99.9%) were provided by Sinopharm Chemical Reagent Co. Ltd., Shanghai, China. P(VDF-TrFE) (50:50 mol. %) powder was purchased from Wuhan CYMENES Technology Co. Ltd., Wuhan, China. SnO2 F (FTO) glass (1.6 mm−14 Ω·cm^−2^) was purchased from Wuhan Jingge Co. Ltd., Wuhan, China.

### 2.2. Preparation of TiO_2_ Nanowire Arrays (TNA)

TiO_2_ nanowire arrays were synthesized through a hydrothermal reaction on FTO glass substrates. In a typical procedure, 1 mL tetrabutyl titanate was added into a mixture of 30 mL deionized water and 30 mL HCL (36.0~38.0%) under continuous stirring. A piece of FTO glass (30 mm × 30 mm × 1.6 mm) was ultrasonically cleaned in a solution containing deionized water, isopropanol and ethanol a volume ration of 1:1:1 for 30 min, and placed into a 100 mL autoclave with its conductive side facing down. The solution, which had been stirred until it became transparent, was transferred into the autoclave. The hydrothermal reaction was conducted at 150 °C for 6 h. After the reaction, the TiO_2_ nanowire arrays (TNA) was rinsed with deionized water and dried.

### 2.3. Fabrication of P(VDF-TrFE) Nanofibers–P(VDF-TrFE)/TiO_2_ Nanowire Arrays (PVTNF-PVT/TNA)

To form collosol in different concentrations, 0.5 g P(VDF-TrFE) powder was dissolved in various volume of DMF. After being stirred for 2 h, the polymer solution was spin-coated on the TNA at 700 rpm for 10 s first, and then 1600 rpm for 30 s. The films were dried at 80 °C for 24 h, and a heat treatment at 180 °C for 1 h was applied to make the polymer completely infiltrate the TNA. The composite was named as PVT/TNA.

A short-lasting electrospinning process was carried out on the surface of the films. The precursor solution was prepared by dissolving 0.5 g P(VDF-TrFE) powder into 3 mL DMF and 2 mL acetone under continuous stirring. The operation voltage was fixed at 13 kV while the distance between the needle and receiver was 13 cm. The electrospinning process was only kept for 2 min to gain a thin hydrophobic layer, then the FTO substrate with film and nanofibers was dried at 60 °C overnight. The composite was named as PVTNF-PVT/TiO_2_. The preparation process is shown in Figure 2.

A parameter TiO_2_ Nanowire (TN) is introduced to distinguish different samples, which refers to the ratio of the length of TiO_2_ nanowires and the thickness of the composite. The TN of the composites are shown in Table 1. According to the different thickness of P(VDF-TrFE), the samples can be named as TN-0.769, TN-0.667, TN-0.594 and TN-0.396. In addition, a pure P(VDF-TrFE) spin-casted film without electrospinning process named as TN-0.000, and a TiO_2_ nanowire array on FTO glass named as TN-1.000 were prepared as contrast samples.

### 2.4. Characterization

The surface and cross-section (dipped in liquid nitrogen to cause brittle failure) morphology of the TiO_2_ nanowire arrays and the composite films were observed by a field emission scanning electron microscopy (FESEM, SU8010, Hitachi Ltd., Tokyo, Japan). The TiO_2_ nanowires were characterized by a transmission electron microscopy (TEM, Tecnai G2 F20, FEI Co., Hillsboro, OR, USA). The surface microstructure and roughness factor were characterized by an atomic force microscopy (AFM, Multimode, VEECO Co., Plainview, NY, USA). The crystal and molecular structure were measured by X-ray powder diffraction (XRD, EMPYREAN, PANalytical Co., Almelo, The Netherlands) with Cu Kα radiation within the 2θ range of 10–80°, and Fourier transform infrared Spectrometer (FT-IR; Nicolet5700, Thermo Electron Corporation, Waltham, MA, USA) in a spectral range of 4000–400 cm^−1^, respectively. The initial contact angle of the composite film was characterized by a video-based contact angle measuring device (OCA 20, Dataphysics, Filderstadt, Germany). Platinum electrodes were plated on the upper side of the film through an ion plating process using a mask with 5 mm diameter holes. Then the dielectric properties measurements were carried out on an impedance analyzer (TH2839, Tonghui, Changzhou, China). The thickness of the composite was measured by a stylus surface profiler (DEKTAK-XT, Bruker, Karlsruhe, Germany).

## 3. Results and Discussions

### 3.1. Surface and Cross-Section Morphology

The SEM images of TNA, PVT/TNA and PVTNF-PVT/TNA are shown in Figure 3. The highly oriented TiO_2_ nanowires appeared to be in the form of pillars with squire sections, with plenty of step profiles on their tops. The average height and diameter of the nanowires were about 2 μm and 150 nm, respectively. According to Figure 3a,b, most of the nanowires were vertical to the FTO substrate, although there was no TiO_2_ seed layer under the arrays. It can be ascribed to the small lattice mismatch between rutile TiO_2_ and FTO and the steric hindrance effect, resulting in the preferred growth orientation on vertical direction [28,29]. The interspace among nanowires was extremely narrow, which means it might be difficult for the polymer to occupy all the space without any void because of the surface tension. The TEM image shown in Figure 6f further confirmed the main crystal face of the TiO_2_ nanowires was (110), with the interplanar spacing of 0.340 nm, matching the rutile TiO_2_.

In addition, Figure 3c,d show the surface and cross-section of PVT/TNA composite. The surface of composite was relatively smooth. No visible flaws could be found and the surface was extremely dense, implying the great quality of the composite film. As shown in Figure 3d, the PVT/TNA composite after heat treatment at 180 °C was dense on its cross-section. P(VDF-TrFE) polymer fused and flowed into the interspace among TiO_2_ nanowires, completely wrapped the TiO_2_ nanowires and formed a well-bonded interface with strong adhesion, contributing to higher dielectric constant and lower dielectric loss.

Furthermore, the SEM image of PVTNF-PVT/TNA shows that P(VDF-TrFE) nanofibers with an average diameter of 200 nm were distributed randomly on the surface of the composite via electrospinning. The nanofibers were interlaced together, and some of them were half embedded in the P(VDF-TrFE) matrix because the DMF solvent still existed in the nanofibers when they reach the surface of P(VDF-TrFE) and integrated with it. The integration also indicates good combination between the electrospinning layer and PVT/TNA composite film. Importantly, nanofibers are piled above another within several tiers, which forms a rough surface and leads to higher roughness and contact angle.

### 3.2. FTIR and XRD Phase Analysis

Figure 4 shows the X-ray diffraction (XRD) patterns of the composite films. Characteristic peaks belonged to FTO substrate can be easily picked out by comparing with the XRD patterns of bare FTO. In detail, peaks at 36.04°, 54.26° and 62.75° are well indexed to the reflections of (101), (211) and (002) planes of rutile TiO_2_ (PDF#01-1292, tetragonal rutile TiO_2_, 0.4594 nm × 0.4594 nm × 0.2959 nm). A sharp peak around 18.89° belongs to the β crystalline phase of P(VDF-TrFE), which indicates the ferroelectricity of the polymer [30,31]. Poly(vinylidene fluoride), also known as PVDF, is a kind of ferroelectric polymers that one hydrogen atom connected to each carbon atom of polyethylene is replaced by a fluorine atom. For PVDF, the β phase means that the fluorine atoms are distributed on the same side of the carbon chain. On this condition, the PVDF molecule possesses larger electric dipole moment, and thus shows greater polarity and dielectric constant. P(VDF-TrFE), a copolymer of VDF and TrFE (containing 20–50 mol.% TrFE), shows a natural preference to form β-phase without any other processes. The 18.89° peak indicates that phase shift did not happen on P(VDF-TrFE) after composing with TiO_2_ nanowire arrays and the heat treatment.

The β-phase P(VDF-TrFE) was then further confirmed by the FTIR patterns, as shown in Figure 5. Peaks belonging to β-P(VDF-TrFE) at 1400 cm^−1^, 1290 cm^−1^ and 840 cm^−1^ can be easily found [32]. Meanwhile α-phase peaks at 764 cm^−1^, 795 cm^−1^ and 976 cm^−1^ do not exist, meaning that the polymer is in complete β-phase. With the TN of the composites decrease, the β-phase peaks become sharper.

### 3.3. Surface Roughness and Contact Angle

The AFM images (Figure 6) show that the roughness of the electrospinning processed surface of composite TN-0.769 was larger than P(VDF-TrFE) membrane (average roughness was 3.63 nm and maximum roughness was 38.1 nm, shown in Figure 6e,f). As shown in Figure 6a,b, with only one tier of nanofiber on the surface, the average roughness was 9.33 nm in a 5 μm × 5 μm area, and the maximum roughness was 113 nm. The maximum roughness was smaller than the diameter of P(VDF-TrFE) nanofibers, proving that the nanofibers were half fused with P(VDF-TrFE) membrane. With interlaced nanofibers covering the smooth P(VDF-TrFE) surface by two tiers (Figure 6c,d), the maximum roughness came to 151 nm.

To obtain a wider variation range of contact angle when a variable voltage is applied between conductive droplet and bottom of the dielectric film, a large initial contact angle is quite necessary. As can be seen in Figure 7, the static contact angle of pure P(VDF-TrFE) membrane with a smooth surface was less than 90°, showing relatively low hydrophobicity. After the electrospinning process was dealt on P(VDF-TrFE), the contact angle increased to about 137°, marking a great reinforcement on hydrophobic property [33]. The composite TN-0.396–TN-0.769 show a similar surface contact angle, indicating that the electrospinning provides a stable promotion on the contact angle. The nanofibers on the surface built a microrough structure that when a droplet was placed on the film, air was sealed between the liquid and the protuberances under the droplet. The wetting type of this structure was the Cassie-Type. According to the Cassie equation, the existence of a liquid–gas interface among a solid–liquid interface significantly increases the contact angle of the membrane to more than 137°, and such a high initial contact angle provides a large range of contact angle for the substantial electrowetting response.

### 3.4. Dielectric Properties

Figure 8a shows the frequency-dependent permittivity of PVTNF-PVT/TNA composite films at room temperature. The dielectric constant of composite TN-0.769 was about 53 at 1 kHz, which is nearly 3 times higher than that of pure P(VDF-TrFE). Permittivity shows a tendency that as the frequency increases, the dielectric constant gets smaller. Additionally, with the thickness of the composite increases, the permittivity of films reduces. For a medium in the alternating electric field, different polarization effects, including electron polarization, dipole polarization, ion polarization and space charge polarization, contribute to the dielectric constant jointly. The high permittivity in low frequency might be explained as the highly aligned TiO_2_ nanowire fillers on the vertical direction, which provides more oriented dipoles distributed along the electric field and thus leads to higher intensity of dipole polarization. Besides, the morphology of TiO_2_ nanowire arrays indicates large and orderly distributed interface with polymer, contributing to the Maxwell–Wagner–Sillars (MWS) polarization, a spatial charge polarization, which refers to the higher concentration of charges on the interface between two different phases due to the difference in electric conductivity. The interface in the composite provides path along the electric field for space interface charges, and contributes to the high dielectric constant [34,35,36]. The β-phase of the P(VDF-TrFE) matrix with a higher electric dipole moment than the α-phase also contributes to the high permittivity. The various volume fractions of TiO_2_ fillers in the composite lead to the relation between dielectric constant and the thickness of the films, because the dielectric constant of TiO_2_ is higher than that of the polymer matrix. In addition, higher volume fraction of TiO_2_ also means larger specific interface between polymer matrix and TiO_2_ fillers, thus the contribution of MWS polarization becomes larger. As the frequency increases, the dipole polarization and space charge polarization are hysteretic to the change of electric field, and thus electron polarization and ion polarization play a more major role on permittivity, which explains the decrease of the dielectric constant.

Moreover, the dielectric loss (Figure 8b) of the composite shows a different pattern with permittivity when the frequency of the alternating voltage changes. Under an alternating electric field, the permittivity of dielectric medium is a complex number, and dielectric loss, also shown as tanδ, is defined as the ratio of its real and imaginary part. The dielectric loss is mainly contributed to the electric heat caused by leak current, while under alternating electric field several kinds of relaxation polarizations became the main factor. As frequency grows, the contribution of electron polarization increases, contribution of leak current and interfacial polarization decreases, and thus the dielectric loss shows a rise tendency [37]. At 1 kHz, the dielectric loss of composite TN-0.769 is about 0.07, only a little bigger than pure P(VDF-TrFE). Generally speaking, the existence of inorganic fillers would obviously raise the dielectric loss of polymer for bringing in the interface and inevitable cavities between two phases. This can explain the rise of dielectric loss when f of the composite increase. The 180 °C heat treat making P(VDF-TrFE) melt helps the polymer and TiO_2_ nanowires form dense interface and the space among nanowire arrays is completely filled with polymer, making the dielectric loss remains in an acceptable range.

Furthermore, Figure 8c shows the curves of frequency-dependent electric conductivity for composite TN-0.769, TN-0.667, TN-0.594, TN-0.396 and TN-0.000. The conductivity of composites obviously decreased with the increase of coating PVT thickness at 10^3^Hz, from 2.24 × 10^−10^ to 2.69 × 10^−9^, indicating that the A composite film possessed great insulation. The low conductivity can effectively prevent the happening of electrolysis. The morphology of the TiO_2_ nanowire arrays determined the good dispersibility of the inorganic fillers and may contribute to the low electric conductivity. Similar to the tendency of dielectric loss electric conductivity get bigger with higher frequency. This is the typical capacitance behavior. In addition, the logarithm of conductivity is approximately linear to the logarithm of frequency, which accords with the Jonscher’s power law [38]:(3)$$\sigma \left(\omega \right)=\sigma_{\mathrm{DC}}+{A\omega }^{n}\;$$
where ω = 2πf.

## 4. Conclusions

In summary, P(VDF-TrFE)/TiO_2_ nanowire arrays composite films have been fabricated through a one-step hydrothermal reaction and a spin-coating process, and then its surface was modified by electrospinning. The surface and cross-section morphology of the composite was studied through SEM to confirm the good orientation of the TiO_2_ nanowires. XRD and FTIR patterns were measured to characterize the phase of TiO_2_ and P(VDF-TrFE). The surface roughness was tested by AFM, and the surface contact angle of the composite rose extremely from 89 to 137° after the electrospinning process, marking the tremendous reinforcement of hydrophobicity. The dielectric properties of the composites were measured. Typically, the permittivity of composite TN-0.769 is 53 at 1 kHz, and the dielectric loss tangent and conductivity were 0.07 and 2.69 × 10^−9^ S/cm, respectively. The great increase in permittivity from pure P(VDF-TrFE) (15 at 1 kHz) could be attributed to the β-phase of P(VDF-TrFE) matrix and the well aligned dipoles and interface along electric field, which could strongly enhance the interfacial polarization. The advantages in both dielectric properties and hydrophobicity demonstrated the great potential of those P(VDF-TrFE)/TiO_2_ nanowire arrays based composites in the application of electrowetting devices.

## Figures and Tables

**Figure 1 polymers-13-00105-f001:**
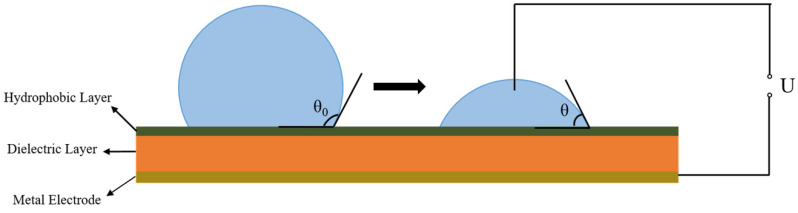
Typical structure of an electrowetting on dielectric (EWOD) device and the change in shape of droplets when voltage is applied.

**Figure 2 polymers-13-00105-f002:**
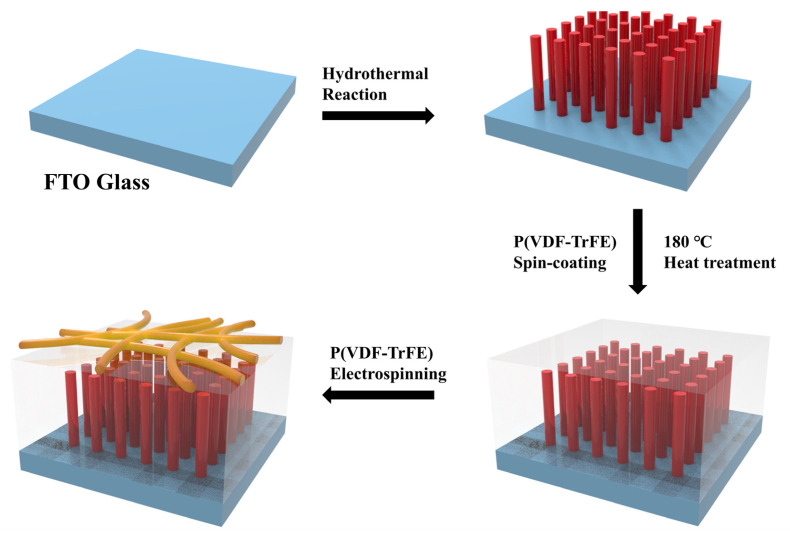
The fabrication process of the PVTNF-PVT/ TiO_2_ nanowire arrays (TNA) composite.

**Figure 3 polymers-13-00105-f003:**
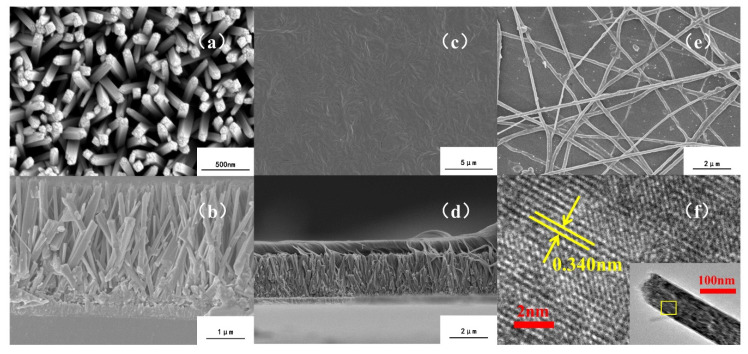
Surface and section-cross SEM images of (**a**,**b**) TNA; (**c**,**d**) PVT/TNA; (**e**) PVTNF-PVT/TNA and (**f**) the TEM image of a single TiO_2_ nanowire.

**Figure 4 polymers-13-00105-f004:**
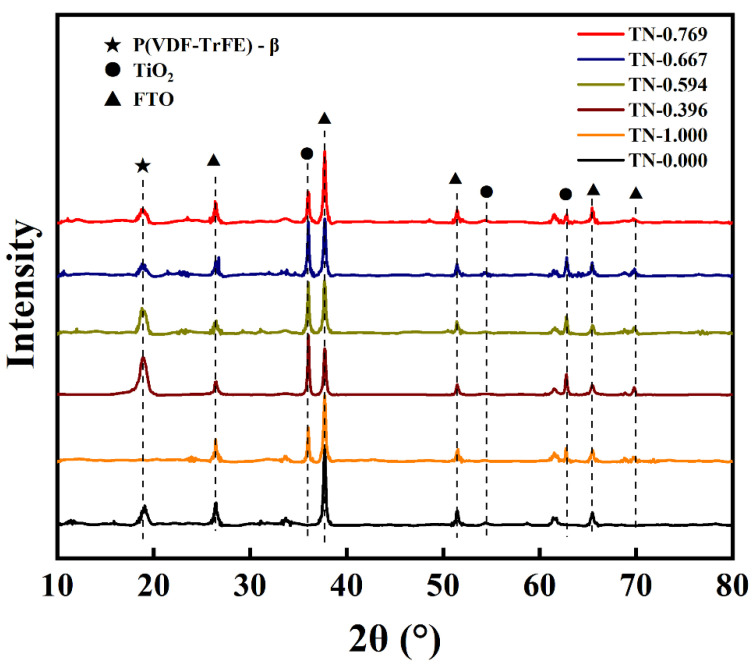
XRD patterns of composite TN-0.769, TN-0.667, TN-0.594, TN-0.396, TN-0.000 and TN-1.000.

**Figure 5 polymers-13-00105-f005:**
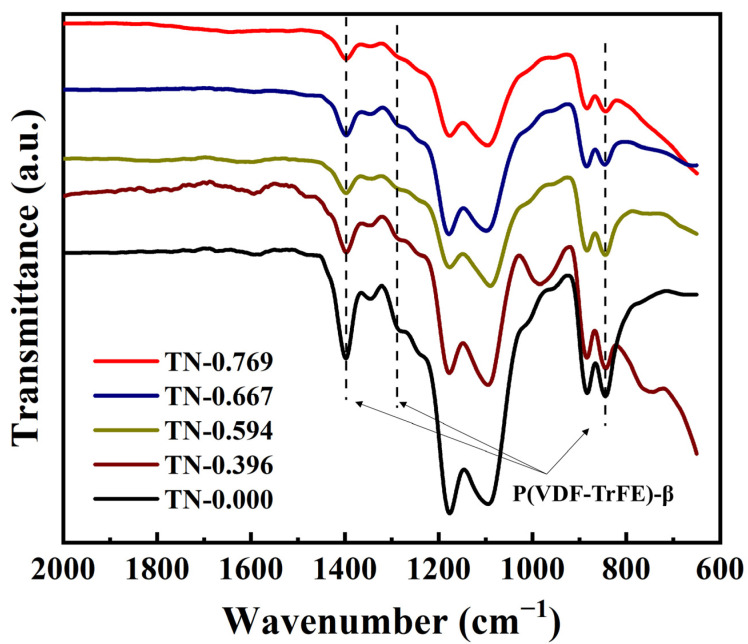
FTIR patterns of composite TN0.769, TN-0.667, TN-0.594, TN-0.396 and TN-0.000.

**Figure 6 polymers-13-00105-f006:**
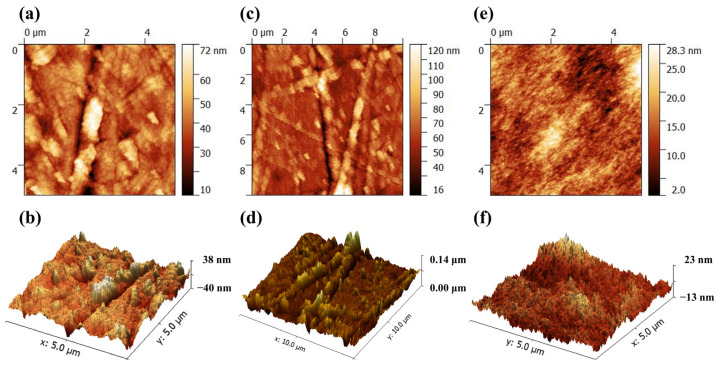
Atomic force microscopy (AFM) images of (**a**–**d**) composite TN-0.769 and (**e**,**f**) TN-0.000.

**Figure 7 polymers-13-00105-f007:**
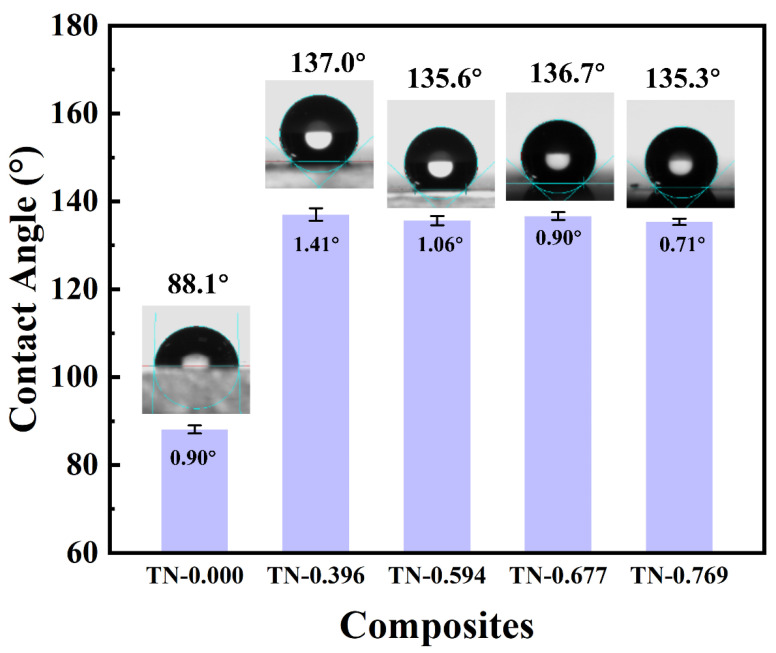
Contact angle of composite TN-0.000, TN-0.396, TN-0.594, TN-0.667 and TN-0.769.

**Figure 8 polymers-13-00105-f008:**
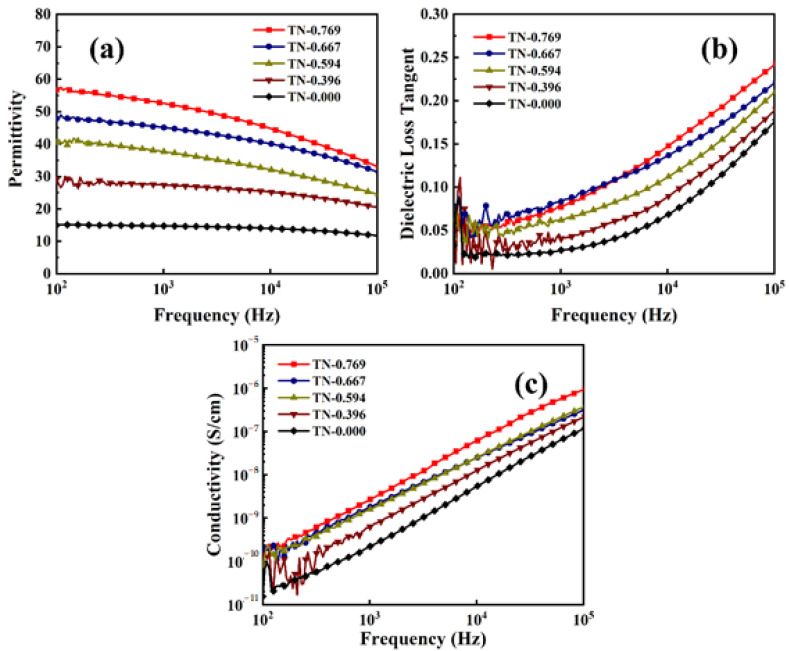
Frequency-dependent (**a**) permittivity, (**b**) dielectric loss and (**c**) electric conductivity of the composite TN-0.769, TN-0.667, TN-0.594, TN-0.396 and TN-0.000.

**Table 1 polymers-13-00105-t001:** The TiO_2_ nanowire lengths, film thickness and TN of composites.

Composites	Length of TiO_2_ Nanowires (μm)	Film Thickness (μm)	TN
TN-0.769	2.0	2.6	0.769
TN-0.667	2.0	3.0	0.667
TN-0.594	1.9	3.2	0.594
TN-0.396	2.1	5.3	0.396
TN-0.000	0	3.2	0.000
TN-1.000	2.1	2.1	1.000

## Data Availability

The data presented in this study are available from the corresponding author on reasonable request.

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
