# Peer review of "Enhanced Dielectric and Hydrophobic Properties of Poly(vinylidene fluoride-trifluoroethylene)/TiO2 Nanowire Arrays Composite Film Surface Modified by Electrospinning"

_polymers, 2020, doi:10.3390/polym13010105_

Round 1
Reviewer 1 Report
The full text is basically descriptive but lacks thorough analysis to the experimental results. This makes the manuscript be like an experimental report rather than an academic paper. The discussion and conclusion needs to be improved
The novelty and essential advance of the present work with respect to the state-of-the-art in the field is not sufficiently apparent in your manuscript. The introduction needs to be improved
The quality of images and figures can be improved, 3,4,6 and 8 figures need more resolution.
Reviewer 2 Report
Dear Authors,
in your interesting manuscript, the following points should be added/changed to further improve it:
- Fig. 2: Does the spin-coated material completely cover the nanowires, or in other words, are TiO2 nanowires and electrospun nanofibers in contact?
- Below (and directly correlated): Do all nanowires have identical lengths so that a TN value < 1 means that there is no contact between nanowires and electrospun nanofibers / air? If possible, give standard deviations in Table 1.
- 3.1: steric hindrance
- 3.2: peaks at 36.04 ... - unit?
- Fig. 4: Please make the inset descriptions larger, e.g. by placing them beside the graph. Please enlarge the axis numbers for all figures.
- Fig. 5: Please add the missing spaces between physical properties and brackets around the units. Above: What is "f"?
- Fig. 6: Please insert scale bars (simply by Gwyddion or the like).
- Fig. 7: Connecting dots by lines is only allowed for equidistantly or logarithmically or ... spaced x-axes. Please use dots or bars. Besides, please add the error bars. And TN-0.000 is no defined in Table 1.
- Fig. 8: All Axis labels and insets are much too small to be readable.
- Generally, your study aims at producing proper samples for electrowetting. So - where are the electrowetting properties?
